

# Light sterile neutrinos: a critical overview

**Ivan Esteban**[1][⋆]

**1** Departament de Física Quàntica i Astrofísica and Institut de Ciències del Cosmos, Universitat de Barcelona, Diagonal 647, E-08028 Barcelona, Spain

⋆ ivan.esteban@fqa.ub.edu

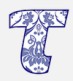

## Abstract

**Light sterile neutrinos, first proposed after the results of the LSND experiment, have been a polemical topic for the last decades after seemingly contradictory data appeared from different experiments. In this overview, I review the experimental hints that point towards sterile neutrinos as well as their statistical compatibility. Even though the muon neutrino disappearance experiments strongly rule out vanilla sterile neutrinos, each oscillation channel remains internally mostly consistent. In any case, in the near future a series of independent and precise experiments should finally settle down this issue.**

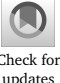
# 1 Introduction

Neutrinos are an essential part of the Standard Model of particle physics (SM). Originally, they were introduced minimally, just to explain their interactions: due to the chiral structure of the SM, only left-handed neutrinos were required. Because of that, the SM has an accidental global flavour symmetry $U(1)_{L_e} \times U(1)_{L_\mu} \times U(1)_{L_\tau}$: each of the leptonic flavours $L_\alpha$ is separately conserved, total lepton number is conserved as well, and as a consequence neutrinos are strictly massless [1, 2].

For the last decades, however, different experiments have accumulated data that conclusively shows that neutrinos change their leptonic flavour after travelling for long distances (see Ref. [3] for an overview). That is, there is conclusive evidence for new physics beyond the SM in the leptonic sector. The minimal way of explaining these flavour transitions is by giving neutrinos a mass, that, as in the quark sector, opens the possibility of flavour mixing and oscillations [4, 5]. In general, for $N$ light neutrino mass eigenstates, the charged current leptonic interaction Lagrangian reads

$$-\mathcal{L}_{\mathrm{CC}} = \frac{g}{\sqrt{2}} W_\mu^+ \sum_{\substack{\alpha \in \{e, \mu, \tau\} \\ i \in \{1, \dots, N\}}} U_{\alpha i} \bar{l}_\alpha \gamma_\mu \nu_i \,, \tag{1}$$

where $U$ is a $3 \times N$ mixing matrix — we know from LEP that there are only three light neutrino flavours that couple to electroweak gauge bosons [6].

As an experimental consequence, charged current interactions will produce and detect superpositions of neutrino mass eigenstates. Furthermore, if the mass eigenstates have different masses, they will evolve differently with time. Therefore, after a neutrino originally produced in a flavour $\alpha$ travels for a distance $L$, the mass eigenstates will interfere and there will be a non-zero probability of observing the neutrino in a flavour $\beta \neq \alpha$. If the neutrino beam travels in vacuum and there, this probability is given by

$$P_{\alpha\beta} = \delta_{\alpha\beta} - 4 \sum_{i<j}^{N} \mathrm{Re}\left[ U_{\alpha i} U_{\beta i}^* U_{\alpha j}^* U_{\beta j} \right] \sin^2 \frac{\Delta m_{ij}^2 L}{4E} + 2 \sum_{i<j}^{N} \mathrm{Im}\left[ U_{\alpha i} U_{\beta i}^* U_{\alpha j}^* U_{\beta j} \right] \sin \frac{\Delta m_{ij}^2 L}{2E} \,, \tag{2}$$

where $E$ is the neutrino energy, $\Delta m_{ij}^2 \equiv m_i^2 - m_j^2$ is the squared-mass splitting among the light neutrino mass eigenstates.

The oscillatory dependence of Eq. 2 on $L/E$ is the smoking gun for neutrino masses as an explanation for neutrino flavour transitions. Precise spectral measurements have led to an accurate observation of two different characteristic frequencies [7–9]. That is, numerous experiments have independently measured two different squared-mass splittings ($\mathcal{O}(10^{-3}\mathrm{eV}^2)$ and $\mathcal{O}(10^{-5}\mathrm{eV}^2)$), which point to three light neutrino mass eigenstates.

# 2 $\overset{(-)}{\nu}_\mu \to \overset{(-)}{\nu}_e$: hints towards a fourth light neutrino mass eigenstate?

## 2.1 LSND

The so-called $3 \times 3$ paradigm (3 leptonic flavours and 3 light neutrino mass eigenstates) to explain the observed neutrino flavour transitions is well established. There exist, however, some discrepancies from this framework, the first of which came from the LSND experiment in the 90s [10]. This experiment had a well-understood neutrino source: a beam of protons hit a target, producing pions. The $\pi^-$ were mostly absorbed, whereas the $\pi^+$ decayed at rest

$$\pi^+ \to \mu^+ + \nu_\mu \,, \tag{3}$$

and, finally, the $\mu^+$ again decayed at rest

$$\mu^+ \to e^+ + \nu_e + \bar{\nu}_\mu, \tag{4}$$

producing a very monochromatic beam of muon antineutrinos. After travelling for about 30 m, any electron antineutrino to which the muon antineutrinos could have transitioned was detected through inverse beta decay

$$\bar{\nu}_e + p \to n + e^+. \tag{5}$$

Surprisingly, the LSND collaboration reported a $3.8\sigma$ $\bar{\nu}_e$ excess over background. This excess, shown in Fig. 1, was interpreted as due to mass-induced neutrino flavour oscillations. If this were the case, the typical LSND $L/E$ ratio requires, from Eq. 2, a squared-mass splitting $\Delta m^2 \sim \mathcal{O}(\text{eV}^2)$. This splitting is several orders of magnitude larger than the other two well-established ones, and therefore explaining LSND through massive neutrino oscillations requires a fourth light neutrino mass eigenstate. Due to unitarity, this means there must exist a fourth neutrino flavour eigenstate that, due to LEP constraints [6], cannot couple to the $Z$ boson. That is, if interpreted as due to neutrino oscillations, LSND points towards a sterile neutrino.

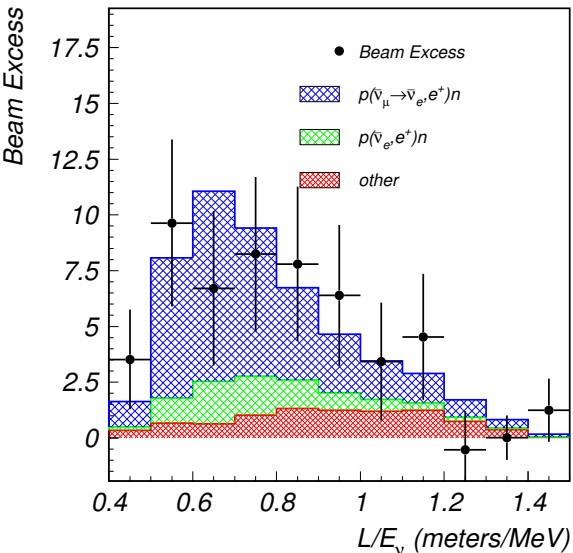

Figure 1: $\bar{\nu}_e$ excess over background (green) observed by the LSND experiment. The blue line corresponds to the prediction if the excess is due to neutrino masses with $\Delta m^2 \sim \text{eV}^2$

The LSND results are, however, rather polemical: an independent reanalysis reevaluated the neutrino fluxes, backgrounds and systematics; lowering the significance of the excess to $2.3\sigma$ [11, 12].

## 2.2 MiniBooNE

The MiniBooNE experiment was built to independently confirm or falsify the LSND signal. Both production and detection were different from LSND: to produce a $\overset{(-)}{\nu}_\mu$ beam, a beam of protons hit a target, producing pions that decayed in flight to either $\nu_\mu$ or $\bar{\nu}_\mu$, depending on the experimental setup. The detector, that could detect both $\overset{(-)}{\nu}_\mu$ and $\overset{(-)}{\nu}_e$, was both a Cerenkov and scintillation detector. Furthermore, the baseline of the MiniBooNE experiment was about

one order of magnitude larger than the LSND one, even though both experiments operated at similar values of $L/E$.

The results from this experiment were, however, intriguing. The observed $\nu_e$ and $\bar{\nu}_e$ spectra are shown in Fig. 2: the $\bar{\nu}_e$ channel shows an excess that looks quite compatible with LSND. In the $\nu_e$ channel, though, the excess at the $L/E$ region explored by LSND is not that significant. What is more, both channels show a low-energy excess that, particularly for the $\nu_e$ channel, is difficult to accommodate with mass-induced neutrino flavour oscillations (whose prediction is shown in dashed lines).

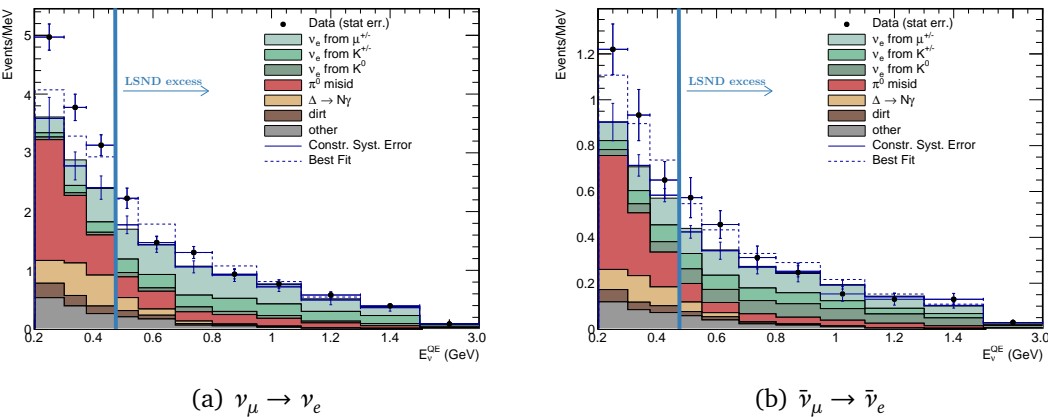

(a) $\nu_\mu \to \nu_e$          (b) $\bar{\nu}_\mu \to \bar{\nu}_e$

Figure 2: MiniBooNE excess in the $\nu_e$ and $\bar{\nu}_e$ channels. The backgrounds are shown in solid lines, whereas the best fit assuming oscillations is the dashed line. The region of $L/E$ where LSND saw the excess is located to the right of the vertical blue line.

## 2.3   Combination of $\overset{(-)}{\nu}_e$ appearance experiments

Even though both LSND and MiniBooNE point towards neutrino flavour violation and therefore new physics, their interpretation in terms of a light sterile neutrino is unclear. On the one hand, the low-energy MiniBooNE excess does not look like oscillations at all; on the other hand, one also has to consider null results from other $\overset{(-)}{\nu}_\mu \to \overset{(-)}{\nu}_e$ experiments. A complete answer, thus, requires a global fit. Fig. 3 shows the combined result of $\overset{(-)}{\nu}_e$ appearance searches from the most recent global fit [13][1]. Since in the $L/E$ regime at short baselines sensitive to $\Delta m^2 \sim \mathcal{O}(\text{eV}^2)$ the other squared-mass splittings are negligible, the $\overset{(-)}{\nu}_\mu \to \overset{(-)}{\nu}_e$ oscillation probability can be parametrised as

$$P\left(\overset{(-)}{\nu}_\mu \to \overset{(-)}{\nu}_e\right) = \sin^2\left(2\theta_{\mu e}\right)\sin^2\left(\frac{\Delta m^2 L}{4E}\right), \tag{6}$$

where $\sin^2(2\theta_{\mu e}) = 4|U_{e4}|^2|U_{\mu4}|^2$, and $\Delta m^2 = \Delta m^2_{41}$. The confidence regions in the global fit are shown in terms of these two parameters.

As can be seen in the figure, all the $\overset{(-)}{\nu}_\mu \to \overset{(-)}{\nu}_e$ experiments are quite consistent. The combined region, shown in red, excludes no flavour transitions with about $6\sigma$: the LSND and MiniBooNE excesses are not a statistical fluctuation. The goodness-of-fit, though, is not that good: Ref. [13] finds $\chi^2_{\min}/\text{dof} = 89.9/(69-2)$, which corresponds to a $p$-value of 3%. This is mostly driven by the MiniBooNE low-energy excess, that as has been mentioned, is rather difficult to fit with oscillations.

---

[1] It does not include the latest MiniBooNE results, but the qualitative conclusions should not change.

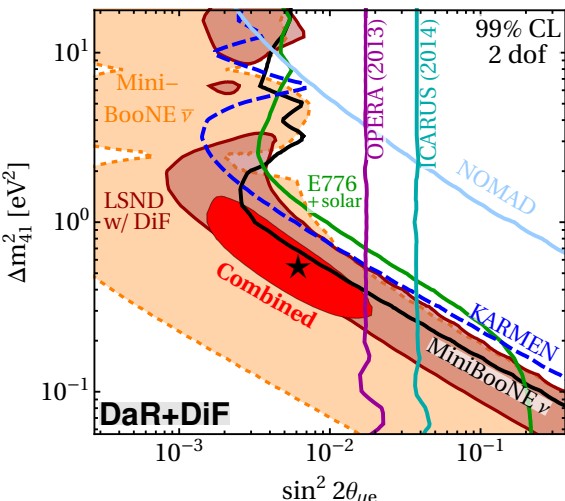

Figure 3: Combined $\overset{(-)}{\nu}_\mu \to \overset{(-)}{\nu}_e$ results. All the parameter space inside the coloured regions is allowed at 99% CL by the two experiments that saw a positive signal: LSND and MiniBooNE. All the parameter space to the right of the coloured lines is disfavoured at 99% CL by experiments that did not see any significant excess above background. The regions are shown in the $\Delta m^2$ vs $\sin^2(2\theta_{\mu e})$ plane: see text for details. Figure extracted from Ref. [13].

# 3 $\overset{(-)}{\nu}_e \to \overset{(-)}{\nu}_e$ disappearance

If the LSND and MiniBooNE signals were due to an eV-scale sterile neutrino, we would not only see neutrino flavour violation in $\overset{(-)}{\nu}_\mu \to \overset{(-)}{\nu}_e$. Instead, there should also be signals in many other short-baseline experiments. And, in fact, another hint in favour of light sterile neutrinos comes from short-baseline reactor experiments.

These hints first started to appear after Refs. [14, 15] re-evaluated the $\bar{\nu}_e$ fluxes from nuclear reactors. Using more sophisticated *ab-initio* calculations and more modern data, they found that the theoretical flux was being overestimated by a factor of about 3%. That is, all the short-baseline reactor experiments, that were not seeing any unusual result, were actually seeing a 3% $\bar{\nu}_e$ deficit. Fig. 4 shows the measured flux normalisation from short-baseline reactor experiments before the re-evaluation: all data points consistently sit below the most recent prediction.

This $\bar{\nu}_e$ deficit can be attributed to short-baseline flavour oscillations. Intriguingly, the $L/E$ ratio points towards $\Delta m^2 \sim \mathcal{O}(\text{eV}^2)$, consistently with the LSND and MiniBooNE signals. However, the reactor experiments in Fig. 4 did not have enough energy resolution to disentangle the $\sin^2 \frac{\Delta m^2 L}{4E}$ modulation of the transition amplitude, which is the true smoking gun for neutrino oscillations. Instead, they could only see the averaged effect, and so this reactor antineutrino anomaly might as well be due to flux mismodellings.

In fact, there are some other experimental hints pointing towards theoretical errors. The RENO experiment first reported a bump in their spectrum, at a neutrino energy $\sim 5$ MeV, that was not predicted by the theoretical calculations [16]. The bump is shown in Fig. 5 as accurately measured by the RENO and Daya Bay experiments. Being a $\bar{\nu}_e$ excess, and not a deficit, it cannot be explained by simple oscillations (even though there are some exotic explanations, see Ref. [17]). Therefore, it sheds some doubt on the validity of the nuclear physics calculations that led to the interpretation of the deficit discussed above as due to sterile neutrinos.

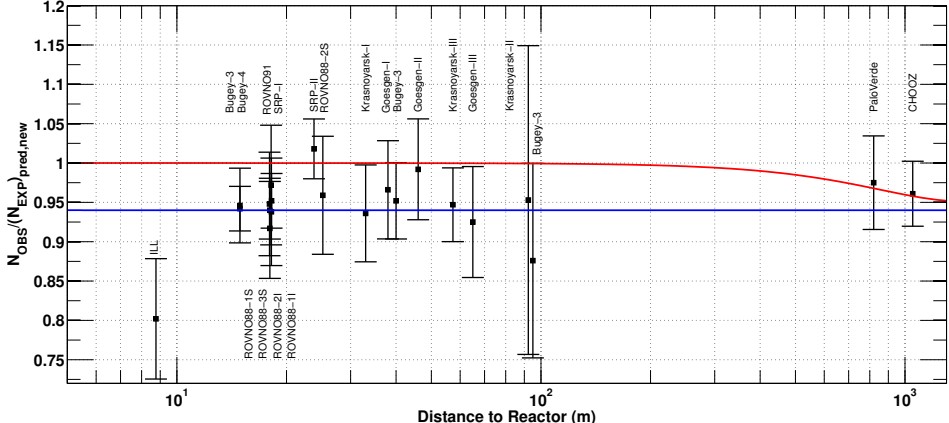

Figure 4: Reactor neutrino flux normalisation as seen in different experiments before the re-evaluation of the fluxes in Refs. [14, 15]. The blue line is the old prediction, the red line is the new one (including the effect due to non-zero $\theta_{13}$). More recent experiments have consistently measured fluxes in accordance with the old calculations.

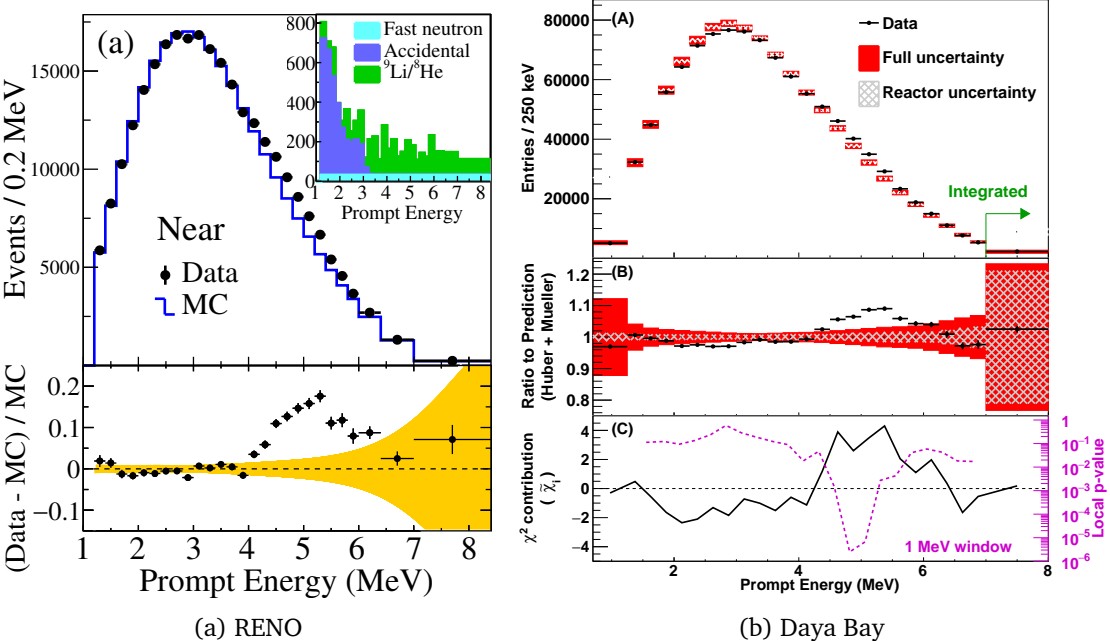

Figure 5: 5 MeV bump as measured by the RENO [16] and Daya Bay [18] collaborations.

## 3.1 Sterile neutrinos?

The interpretation of the reactor antineutrino anomaly as due to sterile neutrinos is supported by some results that are independent of flux calculations. In particular, there are currently two experiments with published data, NEOS [19] and DANSS [20], that have enough energy resolution to disentangle the $L/E$ modulation of the transition probability. The NEOS experiment does it by comparing their results to the experimental flux measured with great accuracy by Daya Bay. The DANSS experiment, on the other hand, has a movable detector that sits close to the nuclear reactor, and they can therefore measure the $L$-dependence of the transition

probability.

Interestingly, both experiments report some "wiggles" in their spectra that can be fitted by sterile neutrino oscillations. Nevertheless, the results, shown in Fig. 6, do not have enough statistical significance yet.

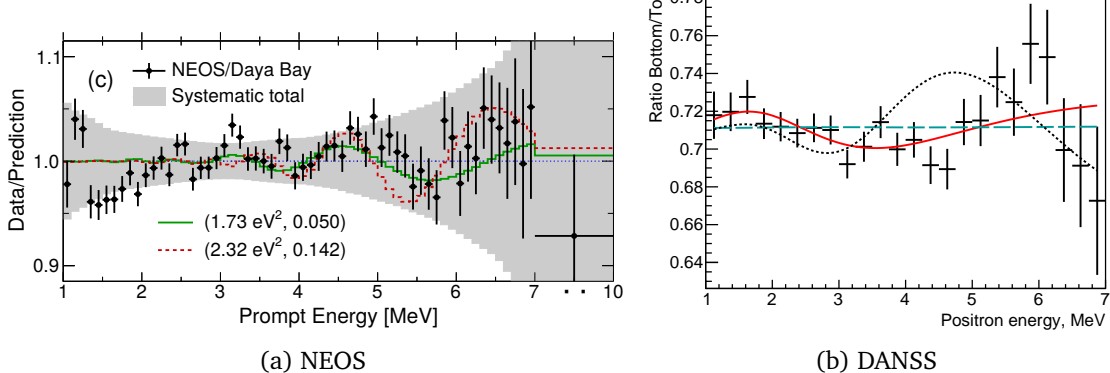

(a) NEOS                      (b) DANSS

Figure 6: Reactor antineutrino spectra measured by the NEOS [19] and DANSS [20] experiments. Both of them see an energy-dependent flavor transition probability that can be fitted by neutrino oscillations, independently of any theoretical flux calculation.

Apart from these results, there were also a set of experiments that reported $\nu_e$ disappearance at short baselines from neutrino sources other than nuclear reactors. In particular, the GALLEX [21] and SAGE [22,23] experiments measured the $^{71}$Ga $\nu_e$ capture cross section with neutrinos coming from intense radioactive sources. They consistently measured a cross section below the theoretical prediction (see Fig. 7), which can be interpreted as due to short-baseline neutrino oscillations. Since the neutrinos came from a radioactive source, these results are completely independent of any theoretical reactor flux calculation.

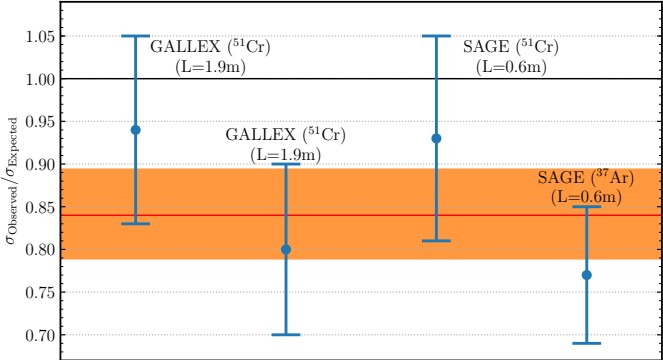

Figure 7: The $^{71}$Ga $\nu_e$ capture cross section as measured by the GALLEX [21] and SAGE [22,23] experiments at short baselines $L$, with neutrinos coming from different radioactive sources (in parentheses). All the data points consistently sit below the theoretical prediction (in black): the measured average and its error are shown in red and orange, respectively. The two GALLEX data points correspond to measurements with two different $^{51}$Cr sources.

## 3.2   Or nuclear physics miscalculations?

Despite the evidences presented in the previous section, there are also experimental results that support flux miscalculations as the origin of the reactor antineutrino anomaly. In more detail, both the Daya Bay [24] and RENO [25] experiments were able to monitor the abundance of the two most common $\beta$-decaying isotopes in their nuclear reactors: $^{239}$Pu and $^{235}$U. Therefore, they could disentangle whether the observed flux deficit affects both isotopes in the same way — the expected result if it were due to sterile neutrino oscillations — or not.

Their results are shown in Fig. 8, where they show the yield (defined as the number of events over the flux) both for $^{239}$Pu and $^{235}$U. Interestingly, their flux measurements for $^{239}$Pu agree with the theoretical expectations from Refs. [14, 15], whereas the $^{235}$U measurements show a deficit with respect to expectations. That is, the results point towards $^{235}$U as the source of the reactor antineutrino anomaly. In principle, this rules out sterile neutrinos as an explanation, because they should affect both isotopes in the same way.

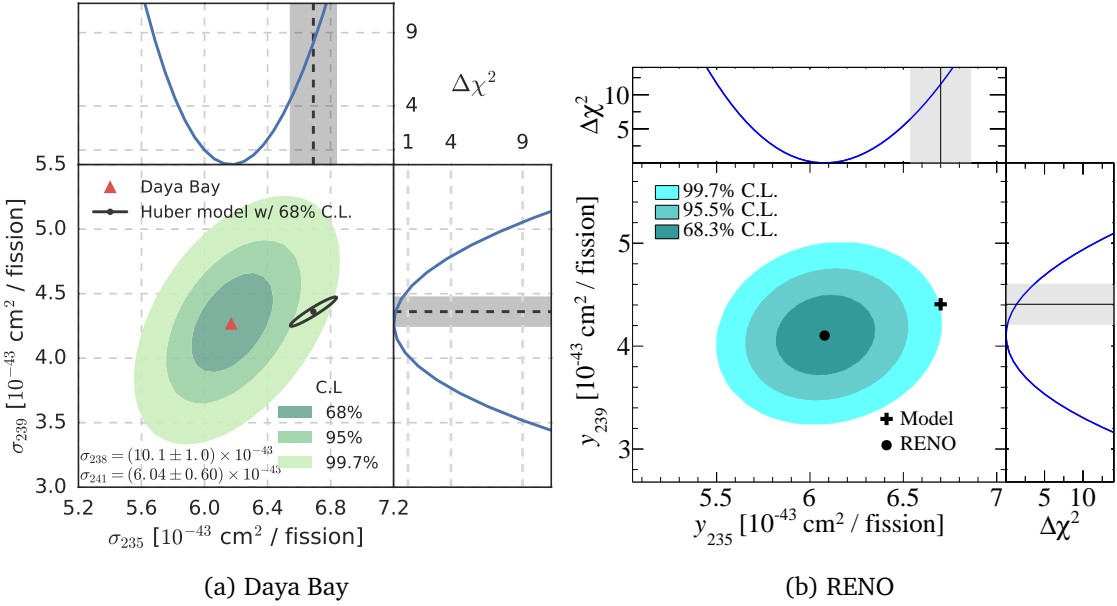

(a) Daya Bay                         (b) RENO

Figure 8: Observed flux yields for $^{239}$Pu, in the vertical axis, and $^{235}$U, in the horizontal axis. Theoretical expectations are shown in black (left) and with a cross (right). The $^{239}$Pu measurement agrees with the theoretical expectations, whereas $^{235}$U shows a deficit. Figures extracted from Refs. [24, 25].

What is more, in Ref. [25], the RENO collaboration showed that the relative amplitude of the 5 MeV bump is correlated with the amount of $^{235}$U in their nuclear reactor. These results again shed doubt on the validity of the theoretical flux calculations for this isotope.

## 3.3   Combination of $\overset{(-)}{\nu}_e$ disappearance experiments

The seemingly contradictory results regarding the origin of the reactor antineutrino anomaly call for a global fit that assesses the compatibility among different data sets. Regarding just reactor results, Ref. [13] combined all the data available in March 2018 under two different hypotheses: they either assumed the theoretical reactor flux calculations from Refs. [14,15], or they left the global normalisation of the reactor flux free (therefore assuming nuclear physics miscalculations).

The results are shown in Fig. 9. Interestingly, both for free and fixed flux normalisations, the 95% CL region points towards the presence of sterile neutrino oscillations. To numerically

quantify the statistical significance of each hypothesis, Tab. 1 shows the minimum $\chi^2$ for either neutrino oscillations or not, and either free or fixed fluxes. Currently, the hypothesis that best describes the data is oscillations and a free flux normalisation: this hypothesis can accommodate both the oscillatory patterns seen by NEOS and DANSS, as well as the isotope-dependent flux measurements. Nevertheless, oscillations and fixed fluxes (i.e., a pure sterile neutrino solution for the reactor antineutrino anomaly) is basically at the same level of confidence as no oscillations and free fluxes (i.e., nuclear physics miscalculations as the only source of the reactor antineutrino anomaly): we still need more data to disentangle this issue.

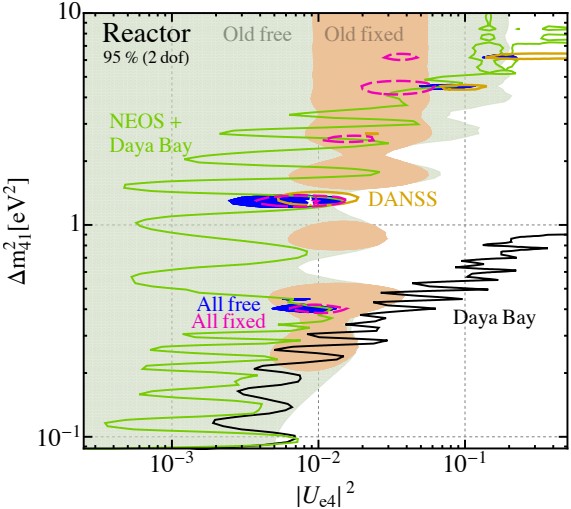

Figure 9: Combined $\bar{\nu}_e$ disappearance results from reactors. All the parameter space inside the coloured regions is allowed at 95% CL, and the parameter space to the right of the lines is disallowed with the same confidence level. The global combination is shown in blue for free flux normalisation, and in pink for fixed theoretical flux calculations. Figure extracted from Ref. [13].

Table 1: $\chi^2_{\min}$ either assuming neutrino oscillations or not, and free or fixed fluxes. Data extracted from Ref. [13].

| Hypothesis | $\chi^2_{\min}$ |
|---|---|
| Oscillations + free fluxes | 185.8 |
| Oscillations + fixed fluxes | 196.0 |
| No oscillations + free fluxes | 197.3 |
| No oscillations + fixed fluxed | 211.5 |

The reactor data can also be combined with the $^{71}$Ga anomaly data, as well as with null searches. The combined regions are shown in Fig. 10: no neutrino oscillations are disfavoured by $3.2\sigma$ ($3.8\sigma$) for free (fixed) reactor fluxes. There is a minor tension between reactor and $^{71}$Ga data, though: the $p$-value for compatibility is 9%, or even 3% once the data from all the other $\overset{(-)}{\nu}_e$ disappearance experiments is included.

## 4 The appearance-disappearance tension and other issues

As shown in the previous sections, there are hints towards the existence of a fourth light neutrino mass eigenstate, coming both from $\overset{(-)}{\nu}_\mu \to \overset{(-)}{\nu}_e$ and from $\overset{(-)}{\nu}_e \to \overset{(-)}{\nu}_e$ experiments.

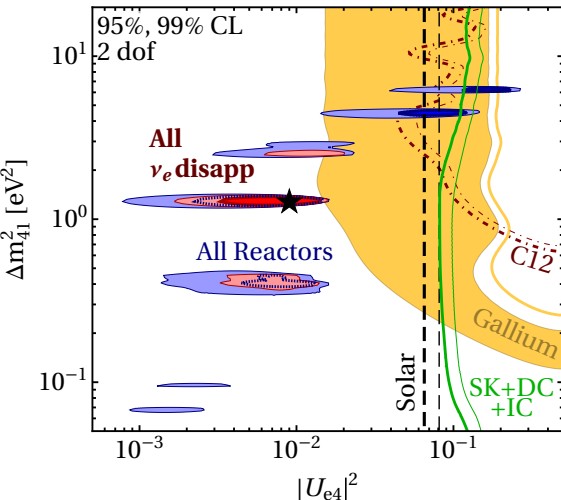

Figure 10: Combined $\overset{(-)}{\nu}_e$ disappearance results, assuming free reactor flux normalisation. All the parameter space inside the coloured regions is allowed, and the parameter space to the right of the solid lines disallowed. Figure extracted from Ref. [13].

Despite some minor internal tensions, they all point in the same direction: a fourth neutrino mass eigenstate with $\Delta m_{41}^2 \sim \mathcal{O}(\text{eV}^2)$, which induces $\overset{(-)}{\nu}_\mu \to \overset{(-)}{\nu}_e$ transitions with an amplitude $4|U_{e4}|^2|U_{\mu4}|^2 \sim 6 \cdot 10^{-3}$, and $\overset{(-)}{\nu}_e \to \overset{(-)}{\nu}_e$ transitions with an amplitude $|U_{e4}|^2 \sim 10^{-2}$.

Therefore, if the picture is consistent, we should also see $\overset{(-)}{\nu}_\mu$ disappearance with an amplitude $|U_{\mu4}|^2 \sim 10^{-1}$. And we do not: Fig. 11 shows the combination of constraints coming from different experiments, that clearly rules out the region allowed by all the data discussed in the previous sections. The global fit in Ref. [13] gives a very low $p$-value for compatibility of $3.71 \cdot 10^{-7}$: vanilla sterile neutrinos as an explanation for the anomalies mentioned in Sections 2 and 3 are ruled out with $4.7\sigma$. This problem is called the *appearance-disappearance tension*.

What is more, the fit does not significantly improve if one single experiment is removed. Not even if we remove all reactor and gallium data, that points towards $|U_{e4}|^2 \sim 10^{-2}$ and therefore gives the required value of $|U_{\mu4}|^2$ to explain LSND and MiniBooNE, does the fit significantly improve.

On top of that, vanilla sterile neutrinos also have severe issues with cosmology. If they are thermally produced in the early universe via mixing, they would increase the relativistic energy content of the universe. The effective number of relativistic degrees of freedom, however, is bound to be [26]

$$N_{\text{eff}} = 3.1 \pm 0.3 \quad (95\%\text{C.L.}), \tag{7}$$

far from $N_{\text{eff}} \sim 4$, the prediction if there is a fourth light neutrino mass eigenstate. Furthermore, massive neutrinos have an imprint on the gravitational potential, and combining CMB and BAO data we get the following bound on the sum of neutrino masses [26]:

$$\sum m_\nu < 0.12\text{eV} \quad (95\%\text{C.L.}), \tag{8}$$

which is hard to reconcile with an eV-scale neutrino mass eigenstate.

Of course, the bounds, coming from cosmology, are indirect. In fact, Refs. [27,28] proposed a way to avoid the cosmological bounds by charging the sterile flavour eigenstate under a new gauge interaction that would suppress the production of the heavy neutrino mass eigenstate in the early universe. These models, however, are currently ruled out [29,30].

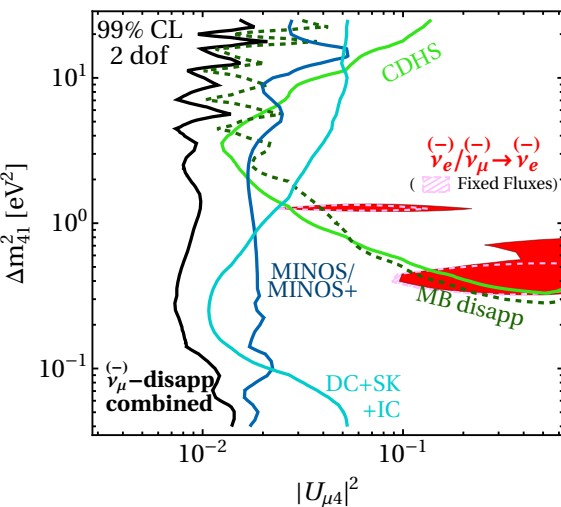

Figure 11: Constraints coming from not observing $\overset{(-)}{\nu}_\mu$ disappearance driven by a fourth light neutrino mass eigenstate. All the parameter space to the right of the solid lines is ruled out with 95% CL. The region preferred at that CL by the $\overset{(-)}{\nu}_\mu \to \overset{(-)}{\nu}_e$ and $\overset{(-)}{\nu}_e \to \overset{(-)}{\nu}_e$ data is shown in red. Figure extracted from Ref. [13].

All in all, models that are free of the appearance-disappearance tension as well as cosmological constraints need to introduce some other new physics in addition to a light sterile neutrino. There are for instance, models that are able to explain the MiniBooNE low energy excess with a very small $|U_{\mu 4}|^2$, and which could also have observable consequences in other experiments [31, 32].

## 4.1 Future prospects

Be that as it may, the final word on the existence of light sterile neutrinos will only come from the experiment. Regarding the $\overset{(-)}{\nu}_\mu \to \overset{(-)}{\nu}_e$ sector, there is a strong short-baseline programme at Fermilab [33] that will start releasing data around 2019. It uses the same beam as MiniBooNE and should be able to confirm or definitely rule out the LSND/MiniBooNE excess. If it is really there, its modern detectors will allow to investigate its origin; and furthermore they have a near detector, unlike MiniBooNE, to better calibrate their backgrounds.

In the $\overset{(-)}{\nu}_e$ disappearance sector, there are many reactor and non-reactor experiments that will continue releasing baseline-dependent and isotope dependent data. If there is a light sterile neutrino behind the reactor antineutrino anomaly, its $L/E$ modulation should definitely show up in the following years.

## 5 Conclusion

There are several hints from neutrino appearance and disappearance experiments at short baselines that point towards new physics in the leptonic sector. These hints come from leptonic flavour violation, a clear signal for physics beyond the Standard Model, with both neutrinos and antineutrinos, different sources, and different detection techniques. They have been mostly interpreted as being due to light sterile neutrinos, even though some of the hints are subject to possible nuclear mismodellings and even though there is a *severe* appearance-disappearance tension. Along with cosmological constraints, the tensions strongly rule out

vanilla sterile neutrinos as an explanation for the anomalies. In any case, in the near future a set of independent and precise experiments will soon settle down this issue, and will point either to uncontrolled experimental uncertainties or to some other exotic new physics.

## Acknowledgements

I would like to thank the organisers of the conference for their invitation and kind hospitality. I would also like to acknowledge Joachim Kopp for helpful discussions.

**Funding information** This work is funded by the FPU program fellowship of the Spanish Ministry of Education, Culture and Sports, FPU15/03697.

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
