# Peer review of "Light sterile neutrinos: a critical overview"

_SciPost Physics Proceedings, doi:SciPost Phys. Proc. 1, 026 (2019)_

## Round 1 · Referee Report · Anonymous · 2018-11-27

Strengths

This paper gives a superb overview over the hotly debated issue of sterile neutrinos. The author very nicely weights all evidence pro and contra. It is well written and clear.

Weaknesses

None.

Report

This paper gives a superb overview over the hotly debated issue of sterile neutrinos. The author very nicely weights all evidence pro and contra. It is well written and clear. I have one question to figure 7: the first two data points are both labelled GALLEX (51CR L = 1.9m), so what is different between these two points?

Requested changes

One typo on line 171: en -> an

  • validity: top
  • significance: top
  • originality: top
  • clarity: top
  • formatting: excellent
  • grammar: excellent

Author:  Ivan Esteban  on 2018-11-28  [id 361]

(in reply to Report 1 on 2018-11-27)

I gratefully thank the referee for his/her careful reading and comments. Answering the question, in Figure 7 the two GALLEX data points refer to measurements with two different 51Cr sources. The particular references for each measurement are

http://inspirehep.net/record/380797

http://inspirehep.net/record/451948

---

## Editorial Decision

published